# Gradient-Free Approaches is a Key to an Efficient Interaction with Markovian Stochasticity

## Abstract

This paper deals with stochastic optimization problems involving Markovian noise with a zero-order oracle. We present and analyze a novel derivative-free method for solving such problems in strongly convex smooth and non-smooth settings with both one-point and two-point feedback oracles. Using a randomized batching scheme, we show that when mixing time $\tau$ of the underlying noise sequence is less than the dimension of the problem $d$, the convergence estimates of our method do not depend on $\tau$. This observation provides an efficient way to interact with Markovian stochasticity: instead of invoking the expensive first-order oracle, one should use the zero-order oracle. Finally, we complement our upper bounds with the corresponding lower bounds. This confirms the optimality of our results.

## 1 Introduction

Stochasticity is a fundamental aspect of many optimization problems, naturally arising in the field of machine learning [48, 28]. Stochastic gradient descent (SGD) [45] and its accelerated variants [39, 25] have become a de facto optimizers for modern large models training. Theoretical properties of SGD have been extensively studied under various statistical frameworks [36, 24, 10, 56], often relying on the assumption that noise is independent and identically distributed (i.i.d.). However, in many real-world applications — including reinforcement learning (RL) [6, 16], distributed optimization [35, 31], and bandit problems [3] — noise is not i.i.d., instead exhibiting correlations or *Markovian structure*.

For instance, in the mentioned growing field of RL, sequential interactions with the environment induce state-dependent structure of the noise, creating a need for non-i.i.d. noise aware algorithms. Although several gradient-based methods for Markovian stochastic oracles have been studied in the past decade [14, 18], policy optimization in RL is based solely on reward feedback, making traditional methods inapplicable, since there is no access to first-order information [46, 9, 19]. *Zero-order optimization* (ZOO) methods are specifically developed to address such problems, and are used in scenarios where gradients are unavailable or prohibitively expensive to compute. Apart from RL, ZOO techniques are widely employed in adversarial attack generation [8], hyperparameter tuning [47, 57], continuous bandits [7, 49] and other applications [54, 33]. While the literature on ZOO is extensive, this work is, to our knowledge, *the first study of optimization problem with both zero-order information and Markovian noise*, aimed at developing an optimal algorithm for a large family of problems from the intersection of these two areas.

Submitted to 39th Conference on Neural Information Processing Systems (NeurIPS 2025). Do not distribute.

## 1.1 Related works

◇ **Zero-order** methods is one of the key and oldest areas of optimization. There are various zero-order approaches, here we can briefly highlight, e.g., one-dimensional methods [32, 42] or their high-dimensional analogues [41], ellipsoid algorithms [58] and searches along random directions [4]. Currently, the most popular and most studied mechanism behind ZOO methods is the finite-difference approximation of the gradient described in [43, 20, 40]. The idea is simple: querying two sufficiently close points is essentially equivalent to finding a value of the directional derivative of the function:

$$\langle \nabla f(x), e \rangle \approx \frac{f(x+te) - f(x)}{t} \approx \frac{f(x+te) - f(x-te)}{2t}, \tag{1}$$

where $e$ is a random direction. It can be a random coordinate, a vector from the Euclidean sphere or a sample of the Gaussian distribution. The approximation (1) in turn leads back to the gradient methods or coordinate algorithms of Nesterov [38]. There are, however, several differences:

• First, to get full gradient information, the algorithm would need $d$ queries instead of one gradient oracle call (here $d$ is the dimension of $x$).

• Second, if the ZO oracle is inexact, i.e. only noisy values of function are available, then finite difference schemes can fail if noise components do not cancel out.

The setting of the second point, when function evaluations experience zero-mean additive perturbations, is called *Stochastic ZOO*. The stochasticity, as noted before, is abundant in the modern optimization world. To tackle this issue, additional assumptions about the noise structure are required. Here we briefly discuss two main ideas adopted in the literature, and refer the reader to Section 2 for precise definitions.

In the case of *two-point feedback*, we assume that for a fixed value of the noise variable one can call the stochastic zero-order oracle at least twice. It means that we can compute the finite difference approximation of the following form:

$$p(x, \xi, e) = \frac{f(x+te, \xi) - f(x-te, \xi)}{2t} \approx \langle \nabla_x f(x, \xi), e \rangle \tag{2}$$

Such approximation produces an estimate for the directional derivative of a noisy realization $f(\cdot, \xi)$ of the function $f$. As mentioned before, the approximation (2) can be used instead of the (stochastic) gradient in first-order methods. In the case of independent randomness, a large number of works are based on this idea. There are results for both non-smooth and smooth convex problems built on classical and accelerated gradient methods of Nesterov and Spokoiny [40]. In the scope of our paper, we are interested in the results for smooth strongly convex problems from [17], namely estimates on zero-order oracle calls to achieve $\varepsilon$-solution in terms of $\|x - x^*\|$: $\mathcal{O}(\frac{d\sigma_2^2}{\mu^2 \varepsilon})$. Here $\sigma_2$ is introduced as the variance of the gradient, i.e. it is assumed that $\mathbb{E}_\xi \nabla f(x, \xi) = \nabla f(x)$ and $\mathbb{E}_\xi \|\nabla f(x, \xi) - \nabla f(x)\|^2 \leq \sigma_2^2$. The main limitation of two-point approach is that several evaluations with the same noise variable are required, which is well suited for problems like empirical risk optimization [34], but can be a major barrier for RL or online optimization.

In the *one-point feedback* setting, a more general stochasticity is assumed. In this case, each call to the zero-order oracle generates a new randomness. Now the approximation (1) looks as follows

$$p(x, \xi^\pm, e) = \frac{f(x+te, \xi^+) - f(x-te, \xi^-)}{2t} \tag{3}$$

Using different $\xi^+$ and $\xi^-$ in (3) renders any conditions on the properties of $\nabla f(\cdot, \xi)$ useless. Instead, it is assumed that $\mathbb{E}_\xi f(x, \xi) = f(x)$ and $\mathbb{E}_\xi |f(x, \xi) - f(x)|^2 \leq \sigma_1^2$. With one-point feedback, the major problem is choosing the right shift $t$ for the finite difference scheme. Picking it too small results in an amplification of the additive noise, and taking $t$ too big leads to a poor gradient estimate. Because of this variance trade-off, the optimal rate for methods with one-point approximation is worse than for two-point feedback. In particular, for smooth strongly convex problems we have the following estimate on zero-order oracle calls [23]: $\mathcal{O}(\frac{d^2 \sigma_1^2}{\mu^3 \varepsilon^2})$.

Although zero-order gradient approximation schemes suffer from high variance, there is a surprising property that makes them superior in *non-smooth* optimization [22, 44, 49]. The idea goes back to the 70s and utilizes the fact that

$$\mathbb{E}[e \cdot p(x, \xi^{(\pm)}, e)] = \frac{1}{d}\nabla f_t(x), \text{ where } f_t \text{ is a } smoothed \text{ function, defined as}$$
$$f_t(x) = \mathbb{E}_r\left[f(x + tr)\right] \text{ with } r \sim RB_2^d$$

In fact, it can be shown that $f_t$ is $\frac{\sqrt{d}G}{t}$-smooth if $f$ is $G$-Lipschitz. This makes zero-order approximation a suitable candidate for a stochastic gradient of $f_t$. Optimizing this function with a first-order method produces some solution, but it may not be the optima of $f$ [22]. From this point, there is a game – for small $t$ the functions $f$ and $f_t$ are closer and for big $t$ the function $f_t$ is easier to optimize as it gets smoother.

In more recent works, there have been many improvements in theoretical understanding of ZO methods. The authors consider higher-order smoothness of the underlying function [2], tackle non-convex non-smooth problems [44], take arbitrary Bregman geometry to benefit in terms of oracle complexity [49, 29], and come up with sharp information-theoretic lower bounds to understand computational limits [15, 1]. But none of them consider Markovian stochasticity.

◇ **Markovian first-order methods.** While the literature on stochastic optimization with i.i.d. noise is extensive, research addressing the Markovian setting remains relatively sparse. In our paper, we focus on the most "friendly" type of uniformly geometrically ergodic Markov chains (see Section 2 for precise definitions).

Duchi et al. [14] conducted pioneering work on non-i.i.d. noise, investigating the Ergodic Mirror Descent algorithm and establishing optimal convergence rates for non-smooth convex problems. For smooth problems there were different attempts to get record-breaking estimates on the first-order oracle [12, 11, 59, 18]. Finally, the optimal results were obtained for both convex and non-convex problems in the works of Beznosikov et al. [5], Solodkin et al. [52]. In particular, for smooth strongly convex objectives under Markovian noise the authors give the complexity of the form: $\mathcal{O}(\frac{\tau\sigma_2^2}{\mu^2\varepsilon})$, where $\tau$ is defined as the mixing time of the corresponding Markov chain (see Section 2). Note that these works utilize Multilevel Monte Carlo (MLMC) batching technique, which helps to effectively interact with Markovian noise. We will need this approach as well. Note that it was first considered in Markovian gradient optimization by Dorfman and Levy [13] for automatic adaptation to unknown $\tau$.

◇ **Hypothesis.** The complexity estimate for strongly convex first-order stochastic methods is $\mathcal{O}(\frac{\sigma_2^2}{\mu^2\varepsilon})$ [36, 37]. Lower bounds for the same class of problems and methods show that the result is unimprovable [58]. As mentioned before, the transition from i.i.d. stochasticity to Markovian stochasticity increases the estimate by $\tau$ times. This result is also optimal as shown by Beznosikov et al. [5]. At the same time, going from gradient oracle to zero-order methods adds a multiplier $d$ in the two-point feedback and $d^2/\varepsilon$ in the one-point case. And this estimate is unimprovable as well [1, 15]. The hypothesis arises that the transition to zero-order Markov optimization adds two multipliers at once: $d\tau$ and $d^2\tau/\varepsilon$ for two- and one-point. It is illustrated in the following diagram for two-point feedback:

| | **FO** | | **ZO 2P** |
|---|---|---|---|
| **IID** | $\frac{\sigma_2^2}{\mu^2\varepsilon}$ | $d$ | $d \cdot \frac{\sigma_2^2}{\mu^2\varepsilon}$ |
| | $\tau$ | ? | ? |
| **Mark.** | $\tau \cdot \frac{\sigma_2^2}{\mu^2\varepsilon}$ | ? | $d\tau \cdot \frac{\sigma_2^2}{\mu^2\varepsilon}$ |

## 1.2 Our contribution

Our main contribution is the answer to the hypothesis above: *surprisingly, it is not true.* In more detail:

◇ **Accelerated SGD.** We present the first analysis of Zero-Order Accelerated SGD under Markovian noise, considering both two-point and one-point feedback. Contrary to the expected multiplicative scaling of convergence rates with both dimensionality and mixing time, our analysis reveals a significant acceleration, as presented in Figure 1. It turns out that if $\tau$ is smaller than $d$, our results do not differ at all from the gradient-free methods

with independent stochasticity. The key technique behind this acceleration is described in Section 2.1. The theory is also numerically validated in Section 3.

◇ **Non-smooth problems.** We also consider non-smooth problems with Markovian noise. Using the smoothing technique we come up with a corresponding upper bounds in this case, as shown in Figure 1. The details of these bounds are presented in Appendix A.

Figure 1: Summary of upper bounds. For notation, see Table 1

|  | Smooth | | Non-smooth | |
|---|---|---|---|---|
|  | **IID** | **Markov.** | **IID** | **Markov.** |
| **FO** | $\frac{\sigma_2^2}{\mu^2\varepsilon}$ [45] | $\tau\frac{\sigma_2^2}{\mu^2\varepsilon}$ [5] | $\frac{G^2}{\mu^2\varepsilon}$ [50] | $\tau\frac{G^2}{\mu^2\varepsilon}$ [14][1] |
| **ZO 2P** | $d\frac{\sigma_2^2}{\mu^2\varepsilon}$ [30] | $(d+\tau)\frac{\sigma_2^2}{\mu^2\varepsilon}$ | $d\frac{G^2}{\mu^2\varepsilon}$ [22] | $(d+\tau)\frac{G^2}{\mu^2\varepsilon}$ |
| **ZO 1P** | $d^2\frac{\sigma_1^2}{\mu^3\varepsilon^2}$ [2][2] | $d(d+\tau)\frac{L\sigma_1^2}{\mu^3\varepsilon^2}$ | $d^2\frac{\sigma_1^2 G^2}{\mu^4\varepsilon^3}$ [23] | $d(d+\tau)\frac{\sigma_1^2 G^2}{\mu^4\varepsilon^3}$ |

◇ **Computational efficiency.** First, as noted above, our method gives the same oracle complexity for any $\tau \leq d$. Moreover, if we assume that calling a zero-order oracle is $d$ times cheaper than computing the corresponding gradient, then the gradient method with Markov noise will require resources proportionally to $d \cdot \tau$ — the cost of one oracle call is $d$ and the complexity scales as $\tau$ for the first-order method from Figure 1. At the same time, the resource complexity of our zero-order method is proportional to $d + \tau$.

◇ **Lower bounds.** In Section 2.3 we establish the first information-theoretic lower bounds for solving Markovian optimization problems with one-point and two-point feedback. Our results match the convergence guarantee of our algorithm up to logarithmic factors, showing that the analysis is accurate and no further improvement is possible.

Table 1: Notations & Definitions

| Sym. | Definition | Sym. | Definition |
|---|---|---|---|
| $\|\cdot\|, \langle\cdot,\cdot\rangle$ | Norm, dot product, assumed Euclidean by default | $\varepsilon$ | $\|x - x^*\|^2$ |
| $\mathsf{Z}, \mathcal{Z}$ | Complete separable metric space, its Borel $\sigma$-algebra | $d$ | Problem dimension |
| $\mathsf{Q}$ | Markov kernel on $\mathsf{Z} \times \mathcal{Z}$ | $L$ | Gradient's Lipshitz constant |
| $\mathbb{P}_\xi, \mathbb{E}_\xi$ | Probability, Expectation under initial distribution $\xi$[3] | $\mu$ | Strong convexity constant |
| $\{Z_k\}$ | Canonical process with kernel $\mathsf{Q}$ | $G$ | Function's Lipshitz constant |
| $RB_2^d, RS_2^d$ | Uniform distribution on unit a $\ell_2$-ball, -sphere | $\sigma_1^2$ | $|F(x,Z) - f(x)|^2 \leq \sigma_1^2$ |
| $e$ | Random direction, $e \sim RS_2^d$ | $\sigma_2^2$ | $\|\nabla F(x,Z) - \nabla f(x)\|^2 \leq \sigma_2^2$ |
| $a_n \lesssim b_n$ | $\exists c \in \mathbb{R}$ (problem-independent): $a_n \leq cb_n$ for all $n$ | $\tau$ | Mixing time of $Z$ |
| $a_n \simeq b_n$ | $a_n \lesssim b_n$ and $b_n \lesssim a_n$ | $g, \hat{g}$ | Gradient estimators |
| $T = \tilde{\mathcal{O}}(S)$ | $T \leq poly(\log S) \cdot S$ as $\varepsilon \to 0$ | $f_t(x)$ | $\mathbb{E}_r\left[f(x+tr)\right], r \sim RB_2^d$ |

## 2  Main results

We are now ready for a more formal presentation. In this paper, we study the minimization problem

$$\min_{x\in\mathbb{R}^d} f(x) := \mathbb{E}_{Z\sim\pi}\left[F(x,Z)\right], \tag{4}$$

where $\pi$ is an unknown distribution and access to the function $f$ (not to its gradient $\nabla f$) is available through a stochastic one-point or two-point oracle $F(x,Z)$.

In our analysis, we will use a set of assumptions on the underlying function $f$ and its oracle, starting with smoothness and convexity:

**Assumption 1.** *The function $f$ is $L$-smooth on $\mathbb{R}^d$ with $L > 0$, i.e., it is differentiable and there is a constant $L > 0$ such that the following inequality holds for all $x, y \in \mathbb{R}^d$:*

$$\|\nabla f(x) - \nabla f(y)\| \leq L\|x - y\|.$$

In the two-point feedback setting, we require the following generalization:

**Assumption 1′.** *For all $Z \in \mathsf{Z}$ the function $F(\cdot, Z)$ is $L$-smooth on $\mathbb{R}^d$.*

Note that the uniform 1′ implies 1.

---

[1]The authors consider general convex case. Using standard restart technique, we get the corresponding bound in the strongly convex case.

[2]The noise is assumed to be point-independent.

[3]By construction, for any $A \in \mathcal{Z}$, we have $\mathbb{P}_\xi(Z_k \in A \mid Z_{k-1}) = \mathsf{Q}(Z_{k-1}, A)$, $\mathbb{P}_\xi$-a.s.

**Assumption 2.** *The function $f$ is $\mu$-strongly convex on $\mathbb{R}^d$, i.e., it is continuously differentiable and there is a constant $\mu > 0$ such that the following inequality holds for all $x, y \in \mathbb{R}^d$:*

$$\frac{\mu}{2}\|x - y\|^2 \le f(x) - f(y) - \langle \nabla f(y), x - y \rangle. \tag{5}$$

We now turn to assumptions on the sequence of noise states $\{Z_i\}_{i=0}^\infty$. Specifically, we consider the case where $\{Z_i\}_{i=0}^\infty$ forms a time-homogeneous Markov chain. Let $Q$ denote the corresponding Markov kernel. We impose the following assumption on $Q$ to characterize its mixing properties:

**Assumption 3.** $\{Z_i\}_{i=0}^\infty$ *is a stationary Markov chain on $(\mathsf{Z}, \mathcal{Z})$ with Markov kernel $Q$ and unique invariant distribution $\pi$. Moreover, $Q$ is uniformly geometrically ergodic with mixing time $\tau \in \mathbb{N}$, i.e., for every $k \in \mathbb{N}$,*

$$\Delta(Q^k) = \sup_{z,z' \in \mathsf{Z}} (1/2) \big\| Q^k(z, \cdot) - Q^k(z', \cdot) \big\|_{\mathsf{TV}} \le (1/4)^{\lfloor k/\tau \rfloor}. \tag{6}$$

Assumption 3 is common in the literature on Markovian stochasticity [14, 12, 13, 5, 52]. It includes, for instance, irreducible aperiodic finite Markov chains [18]. The mixing time $\tau$ reflects how quickly the distribution of the chain approaches stationarity, providing a natural measure of the temporal dependence in the data.

Next, we specify our assumptions on the oracle. As discussed in Section 1.1, these assumptions differ based on the type of feedback.

**Assumption 4** (for one-point). *For all $x \in \mathbb{R}^d$ it holds that $\mathbb{E}_\pi[F(x, Z)] = f(x)$. Moreover, for all $Z \in \mathsf{Z}$ and $x \in \mathbb{R}^d$ it holds that*

$$|F(x, Z) - f(x)|^2 \le \sigma_1^2,$$

**Assumption 4′** (for two-point). *For all $x \in \mathbb{R}^d$ it holds that $\mathbb{E}_\pi[\nabla F(x, Z)] = \nabla f(x)$. Moreover, for all $Z \in \mathsf{Z}$ and $x \in \mathbb{R}^d$ it holds that*

$$\|\nabla F(x, Z) - \nabla f(x)\|^2 \le \sigma_2^2.$$

Recent works on stochastic ZOO methods have considered milder assumptions, such as bounded variance (see Section 1.1). However, the uniform boundedness assumed in Assumptions 4 and 4′, is standard in analyses under Markovian noise [14, 12, 13, 5, 52]. These assumptions can be relaxed under stronger conditions, e.g., uniform convexity and smoothness of $F(\cdot, Z)$ [18].

Assumptions 3 and 4 allow us to reduce the variance of the noise via batching, similarly the to i.i.d. setting. This is captured in the following technical lemma:

**Lemma 1.** *Let Assumptions 3 and 4(4′) hold. Then for any $n \ge 1$ and $x \in \mathbb{R}^d$ and any initial distribution $\xi$ on $(\mathsf{Z}, \mathcal{Z})$, we have*

$$\mathbb{E}_\xi\left[ \frac{1}{n}\sum_{i=1}^n F(x, Z_i) - f(x) \right]^2 \lesssim \frac{\tau}{n}\sigma_1^2, \quad \mathbb{E}_\xi\left\| \frac{1}{n}\sum_{i=1}^n \nabla F(x, Z_i) - \nabla f(x) \right\|^2 \lesssim \frac{\tau}{n}\sigma_2^2.$$

## 2.1 Batching technique

In this section, we describe the main tools used to establish the $(d + \tau)$-type scaling of the error rate. We will focus on reducing the variance and bias of gradient estimators using a specialized batching approach.

We begin by fixing a common building block of our gradient estimators at a point $x$ for both one-point and two-point feedback, as introduced in Section 1.1:

$$\hat{g}(x, Z^{(\pm)}, e) = d \cdot p(x, Z^{(\pm)}, e) \cdot e = e \cdot \begin{cases} d\dfrac{F(x + te, Z^+) - F(x - te, Z^-)}{2t} & \text{(one-point)}, \\ d\dfrac{F(x + te, Z) - F(x - te, Z)}{2t} & \text{(two-point)}. \end{cases}$$

These estimators exhibit a twofold randomness that affects how rapidly they concentrate around the true gradient, as we will discuss below.

For clarity, we focus our discussion on the one-point case, although our conclusions extend to the two-point case as well.

A widely used variance reduction technique is *mini-batching*, where one computes $F(x, Z_i)$ over a batch of noise variables $\{Z_i\}_{i=1}^n$. The mini-batch gradient estimator is given by:

$$\hat{g}_{mb}(x) = \frac{1}{n}\sum_{i=1}^n \hat{g}(x, Z_i^\pm, e) = e \cdot d \overbrace{\left(\frac{1}{n}\sum_{i=1}^n p(x, Z_i^\pm, e)\right)}^{p_{mb}}.$$

Let us estimate the scaling of its variance $\mathbb{E}_e\mathbb{E}_Z\|\hat{g}_{mb} - \nabla f\|^2$ with the noise level $\sigma_1^2$. As $E_Z\hat{g}_{mb} \approx d\frac{f(x+te)-f(x-te)}{2t} \approx d\langle\nabla f, e\rangle$ we would like to estimate the following for any fixed direction $e$:

$$\mathbb{E}_Z\big[d \cdot p_{mb}(x) - d\langle\nabla f, e\rangle\big]^2 \approx \frac{d^2}{t^2}\mathbb{E}_Z\Big[\frac{1}{n}\sum_{i=1}^n F(x+te, Z_i^+) - f(x+te)\Big]^2 \overset{(1)}{\approx} \frac{d^2\tau}{n}\frac{\sigma_1^2}{t^2}. \quad (7)$$

With that, we bound the variance:

$$\mathbb{E}_e\mathbb{E}_Z\|\hat{g}_{mb} - \nabla f\|^2 \gtrsim \mathbb{E}_e\mathbb{E}_Z\|\hat{g}_{mb} - \mathbb{E}_Z\hat{g}_{mb}\|^2 \approx \mathbb{E}_e\mathbb{E}_Z\|\hat{g}_{mb} - d\langle\nabla f, e\rangle\|^2 \overset{(7)}{\approx} \frac{d^2\tau\sigma_1^2}{nt^2}. \quad (8)$$

**Can the mini-batching scheme be improved?**
This subsection explores an unexpected source of improvement that contradicts our initial hypothesis. Specifically, we identify an inefficiency in the current use of samples $Z_i$, which becomes evident from two perspectives. Equation (8) shows the variance scales as $\frac{\tau}{n}$. If we could reduce $\tau$ by a factor of $k$, we would need $k$-times fewer samples to maintain the same variance. This leads us to the idea of sparsified sampling. We partition the Markov noise chain $\{Z_i\}$ into $k$ subchains $\{Z_{k\cdot i+r}\}$ for $r = 0\ldots k-1$. This corresponds to a mixing time of $\lceil\frac{\tau}{k}\rceil$ for each subchain (see (3)), effectively reducing temporal correlation - a natural consequence of sampling every $k$-th element of the original chain. Thus, sampling from any single subchain could yield a $\min(k, \tau)$-fold reduction in the number of samples needed (although such procedure would still require all intermediate oracle calls, yielding no computational speedup).

For a concrete illustration of that inefficiency, consider a lazy Markov chain that remains in the same state for (an average of) $\tau$ steps before transitioning uniformly at random. In such a case, all oracle queries $F(x, Z)$ for a fixed $x$ return the same value for $\tau$ consecutive steps. Therefore, retaining only every $\tau$-th estimate $\hat{g}$ would yield a mini-batch of equivalent quality.

In summary, we observe that the mini-batching scheme could, in principle, operate just as effectively by retaining only every $k$-th sample and discarding the rest. This might suggest that better utilization of the samples is possible. First order methods, nevertheless, are unable to exploit this redundancy (as shown by [5]'s lower bound) and are effectively forced to wait out the $\tau$-step mixing window. In contrast, we can exploit this structure by querying finite differences along different directions to estimate the gradient better. Specifically, we construct $d$ subchains, and use the sample from the $r$-th subchain $Z_{d\cdot i+r}$ to estimate $r$-th partial derivative $\frac{F(x+te_r, Z)-F(x-te_r, Z)}{2t}$, effectively restoring the full gradient coordinate-wise.

Let us estimate the resulting variance reduction. First, we achieve a $d$-fold reduction by reconstructing all $d$ gradient coordinates. Second, each coordinate now operates on a chain with mixing time $\lceil\frac{\tau}{d}\rceil$, yielding an additional factor of $\min(d, \tau)$. However, because batches are now split across $d$ coordinates, each batch is $d$ times smaller than before, introducing a factor of $d$ loss. The net variance reduction is therefore $\min(d, \tau)$, and the final scaling becomes $d \cdot \frac{d\tau}{\min(d,\tau)} = d \cdot \max(d, \tau) \simeq d(d + \tau)$.

**Random directions**
This insight can be extended to a simpler yet equally effective method. Instead of assigning directions deterministically, we associate each sample with a random direction $e \in RS_2^d$, forming the estimator:

$$\hat{g}_{rd}[n](x, Z, e) = \frac{1}{n}\sum_{i=1}^n \hat{g}(x, Z_i, e_i).$$

While the above discussion was intuitive, we now outline a more formal approach (see Lemma 5 for details). As lazy Markov chain is effectively equivalent to stochastic i.i.d. $\tau$-point feedback setting, we follow Corollary 2 of [15], who decompose the total variance into two terms:

$$\mathbb{E}\|\hat{g}_{rd} - \nabla f(x)\|^2 \leq 2\mathbb{E}\|\hat{g}_{rd} - \mathbb{E}_e\hat{g}_{rd}\|^2 + 2\mathbb{E}\|\mathbb{E}_e\hat{g}_{rd} - \nabla f(x)\|^2.$$

Each of the two terms individually eliminates one factor from the $d^2\tau$ dependence.

The first term:

$$\mathbb{E}\|\hat{g}_{rd} - \mathbb{E}_e\hat{g}_{rd}\|^2 = \mathbb{E}_Z\mathbb{E}_e\left\|\frac{1}{n}\sum_{i=1}^n \underbrace{[\hat{g}(x,Z_i,e_i) - E_{e_i}\hat{g}(x,Z_i,e_i)]}_{\mathbb{E}_e[\cdot]=0,\text{ independent w.r.t. } e}\right\|^2$$
$$= \frac{1}{n^2}\sum_{i=1}^n \mathbb{E}\|\hat{g}(x,Z_i,e_i) - \mathbb{E}_{e_i}\hat{g}(x,Z_i,e_i)\|^2$$

is independent of $\tau$ since Assumption 4 bounds each term directly.

For the second term, we observe that $\mathbb{E}_e\hat{g}_{rd} = \mathbb{E}_e\hat{g}_{mb}$, and thus the bound involves $\mathbb{E}\|\mathbb{E}_e\hat{g}_{mb} - \nabla f(x)^2\|$. This is crucially different from the $d^2\tau$ dependence that appeared in the mini-batch case, when we considered $\mathbb{E}\|\hat{g}_{mb} - \nabla f(x)^2\|$. Intuitively, the expectation over directions helps recover the full gradient rather than a directional component, thereby reducing variance with respect to $d$.

**Multilevel Monte Carlo**

The estimator $\hat{g}_{rd}$ is not our final construction. While it controls variance, the temporal correlation in noise may introduce significant bias. A well-established approach to mitigating this is MLMC, widely used in the statistical literature [27, 26], and more recently in gradient optimization [13, 5]. Here is our interpretation.

With parameters $J, l, M, B$ from Table 2, $\{Z_i\}$ - $2^J l$ samples from $Z$ and $\{e_i\}$ - random directions we introduce MLMC estimator:

$$\hat{g}_{ml}(x) = \hat{g}_{rd}[l](x) + \begin{cases} 2^J\left[\hat{g}_{rd}\left[2^J l\right](x) - \hat{g}_{rd}\left[2^{J-1}l\right](x)\right], & \text{if } 2^J \leq M \\ 0, & \text{otherwise} \end{cases}$$

$\hat{g}_{ml}$ is our final gradient estimator, with the following guarantees:

**Lemma 2** (for one-point). *Let Assumptions 1, 3 and 4 hold. For any initial distribution[1] $\xi$ on $(\mathsf{Z}, \mathcal{Z})$ the gradient estimates $\hat{g}_{ml}$ satisfy $\mathbb{E}[\hat{g}_{ml}] = \mathbb{E}\left[\hat{g}_{rd}\left[2^{\lfloor\log_2 M\rfloor}l\right]\right]$. Moreover,*

$$\mathbb{E}\|\nabla f_t(x) - \hat{g}_{ml}(x)\|^2 \lesssim \frac{d\|\nabla f(x)\|^2}{B} + \frac{d^2L^2t^2}{B} + \frac{d(d+\tau)\sigma_1^2}{Bt^2},$$
$$\|\nabla f_t(x) - \mathbb{E}[\hat{g}_{ml}(x)]\|^2 \lesssim \frac{d\tau\sigma_1^2}{t^2BM}.$$

One can note that although $\hat{g}_{ml}$ requires, on average, $\mathbb{E}\left[2^J lB\right] = \log_2^2 M \cdot B$ oracle calls, the variance is only reduced by a factor of $B$. In contrast, the bias is reduced significantly - by a factor of $BM$.

## 2.2 Algorithm

We now present the full version of Algorithm 1, which incorporates the gradient estimators discussed in the previous section and uses a slightly modified variant of Nesterov's Accelerated Gradient Descent at its core.

While technically we prove four separate upper bounds covering both one- and two-point feedback under smooth and non-smooth assumptions, they follow the same scheme which we will illustrate in the one-point smooth case.

---

[1]Note that $\hat{g}_{ml}$ (specifically $Z_1$) indirectly depends on the chain's initial distribution. As our algorithm is going to repeatedly call $\hat{g}_{ml}$, next iteration's initial distribution is current iteration's final distribution. This fact makes the estimates correlated. We sidestep this problem by assuming any initial distribution.

Table 2: Parameters of Algorithm 1

| Hyperparameters | | Momentums | | Batch hidden parameters | |
|---|---|---|---|---|---|
| $\gamma$ | Stepsize, $\in (0; \frac{3}{4L}]$ | $\beta$ | $\sqrt{\frac{4p^2\mu\gamma}{3}}$ | $2^J l$ | Batch size. If $2^J l > M$, then 0 |
| $t$ | Approximation step | $\eta$ | $\frac{3\beta}{2p\mu\gamma} = \sqrt{\frac{3}{\mu\gamma}}$ | $J$ | Random, $J \sim \text{Geom}(1/2)$ |
| $B$ | Batch size multiplier | $\theta$ | $\frac{p\eta^{-1}-1}{\beta p\eta^{-1}-1}$ | $M$ | Batch size limit, $M = \frac{1}{p} + \frac{2}{\beta}$ |
| $N$ | Number of iterations | $p$ | See Appendix | $l$ | $(\lfloor \log_2 M \rfloor + 1) \cdot B$ |

Lemma 4 establishes key properties of the smoothed objective function. Lemma 5 provides bounds on the bias and variance of the baseline estimator $\hat{g}_{rd}$. Lemma 2 then quantifies how the MLMC scheme amplifies or reduces these statistics. Finally, in Section C.4, we combine the results of these lemmas to prove the first part of Theorem 1, bounding Algorithm 1's error. By tuning the parameters appropriately, we obtain the following iteration complexity bound:

---

**Algorithm 1** `Randomized Accelerated ZO GD`

1: **Initialization:** $x_f^0 = x^0$; see Table 2.
2: **for** $k = 0, 1, 2, \ldots, N-1$ **do**
3:     $x_g^k = \theta x_f^k + (1-\theta)x^k$
4:     Sample $J_k, \{e_i\}, \left\{ F(x_g^k \pm te_i, Z_i^{(\pm)}) \right\}$
5:     Calculate $\hat{g}^k = \hat{g}_{ml}(x)$
6:     $x_f^{k+1} = x_g^k - p\gamma \hat{g}^k$
7:     $x^{k+1} = \eta x_f^{k+1} + (p-\eta)x_f^k +$
           $+(1-p)(1-\beta)x^k + (1-p)\beta x_g^k$
8: **end for**

---

**Theorem 1.** *Let Assumptions 1 to 4 hold, and consider problem* (4) *solved by Algorithm 1. Then, for any target accuracy $\varepsilon$ and batch size multiplier $B$ (see Tables 1 and 2 for notation), and for a suitable choice of $\gamma, t, p$, the number of oracle calls required to ensure $\mathbb{E}\|x^N - x^*\|^2 \leq \varepsilon$ is bounded by*

$$B \cdot \tilde{\mathcal{O}}\left[ \max\left(1, \frac{d}{B}\right)\sqrt{\frac{L}{\mu}}\log\frac{1}{\varepsilon} + \frac{Ld(d+\tau)\sigma_1^2}{B\mu^3\varepsilon^2} \right] \quad \text{one-point oracle calls.}$$

**Theorem 1′.** *Let Assumptions 1′ to 4′ hold, and consider problem* (4) *solved by Algorithm 1. Then, for any target accuracy $\varepsilon$ and batch size multiplier $B$ (see Tables 1 and 2 for notation), and for a suitable choice of $\gamma, t, p$, the number of oracle calls required to ensure $\mathbb{E}\|x^N - x^*\|^2 \leq \varepsilon$ is bounded by*

$$B \cdot \tilde{\mathcal{O}}\left[ \max\left(1, \frac{d}{B}\right)\sqrt{\frac{L}{\mu}}\log\frac{1}{\varepsilon} + \frac{(d+\tau)\sigma_2^2}{B\mu^2\varepsilon} \right] \quad \text{two-point oracle calls.}$$

**Remark.** The *iteration complexity* of the algorithm, i.e., the number of iterates $x^k$ generated (equal to the oracle complexity divided by $B$), is bound by $\tilde{\mathcal{O}}\left(\sqrt{\frac{L}{\mu}}\log\frac{1}{\varepsilon}\right)$ as the batch size multiplier $B$ goes to infinity. This matches the optimal convergence rates for optimization with *exact* gradients [39].

## 2.3 Lower bounds

Here we present theorems demonstrating that no algorithm can asymptotically outperform Algorithm 1 in the smooth, strongly convex setting with either one- or two-point feedback.

**Theorem 2.** *(Lower bounds) For any (possibly randomized) algorithm that solves the problem* (4)*, there exists a function $f$ that satisfies Assumptions 1 to 4 (1′ to 4′), s.t. in order to achieve $\varepsilon$-approximate solution in expectation $\mathbb{E}\|x^N - x^*\|^2 \leq \varepsilon$, the algorithm needs at least*

$$\Omega\left(\frac{d(d+\tau)\sigma_1^2}{\mu^2\varepsilon^2}\right) \quad \text{one-point or} \quad \Omega\left(\frac{(d+\tau)\sigma_2^2}{\mu^2\varepsilon}\right) \quad \text{two-point oracle calls.}$$

**Remark.** These results assume bounded second moments rather than uniform noise bounds. We explain how to adapt them to our setting, incurring only logarithmic overheads, in Section E.2.

**Discussion.** We now compare our results to existing work. Akhavan et al. [2] analyze a special case of the one-point setting where the noise is independent of the query points. This

aligns with our one-point oracle model and allows i.i.d. sampling as a Markov chain with fixed mixing time $\tau = 1$. The only factor they do not consider is $\sigma_1^2$, which, however, appears in their proof with additional $\mu^2$ factor if used with scaled Gaussian noise. We discuss this further in Appendix E.

In the work of Beznosikov et al. [5], a first-order Markovian oracle is considered, but the hard instance problem is a one-dimensional quadratic function, which makes first-order and zero-order information equivalent. Their result therefore corresponds to the $d = 1$ case in the two-point regime. Duchi et al. [15] provide tight lower bounds for general convex functions under two-point feedback. Their techniques can be extended to the strongly convex case by incorporating a shared quadratic component across the hard instances, as detailed in Appendix E, Theorem 10, yielding the bound we state for the two-point oracle with $\tau = 1$.

Our novel contribution lies in establishing a lower bound that scales as $d\tau$ in the one-point regime for large $\tau$; see Theorem 9. While our analysis relies on classical tools such as multidimensional hypothesis testing, the Markovian structure requires new bound on distances between joint distributions and the use of clipping. Detailed proofs, discussions, and further remarks on clipping appear in Appendix E.

## 3 Experiments

This section empirically supports our theoretical convergence rates and lower bounds, with particular focus on the stochastic component where we claim linear scaling in $d + \tau$ instead of $d\tau$.

**Setup.** Our setup repeats the problem we used to prove the lower bounds (see Appendix E and [51]). We consider a quadratic objective $f(x) = \frac{1}{2}\|x\|^2$ and a two-point Markovian oracle $F(x, Z) = f(x) + \langle x, Z \rangle$. The noise sequence $\{Z_i\}$ is a lazily updated standard Gaussian vector with variance $\sigma_2^2$. Figure 2 illustrates how the optimization error of Algorithm 1 scales with mixing time, problem dimension, and different values of $\sigma_2^2$.

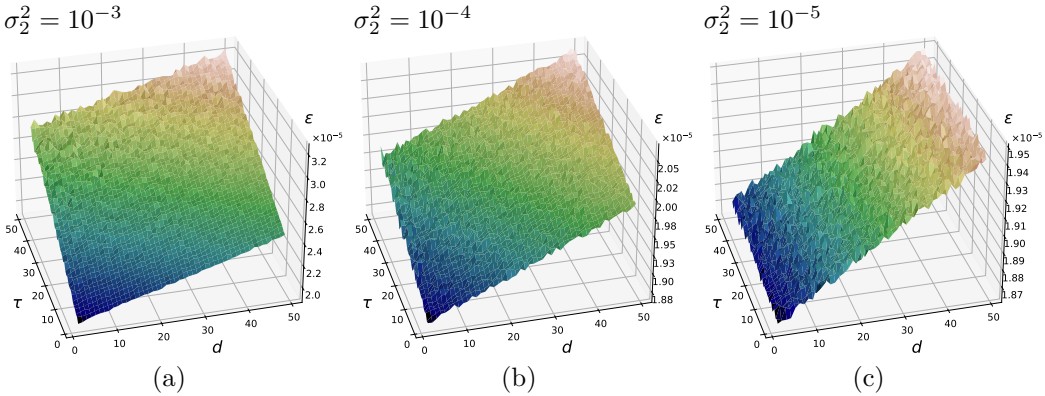

Figure 2: Optimization error $\varepsilon = \|x^N - x^*\|^2$ after $N = 10^3$ iterations. Starting point error $\|x_0 - x^*\|^2 = 10^{-2}$. Stepsize $\gamma = 10^{-3}$, $t = 10^{-5}$. The results are averaged over $10^4$ runs.

**Discussion.** The results confirm the linear dependence of the error on both the problem dimension $d$ and the mixing time $\tau$. The noise parameter $\sigma^2$ controls the influence of the stochastic part. In Fig. (a), where $\sigma_2^2 = 10^{-3}$, the stochastic component dominates, while in Fig. (c), with $\sigma_2^2 = 10^{-5}$, it is negligible. Fig. (b) shows an intermediate regime that smoothly interpolates between the two, yet maintains the linear scaling. The deterministic part (c) shows no dependence on mixing time, but grows linearly with $d$, which aligns with our theory (Theorem 1$'$). The stochastic part (a) scales as $(d + \tau)$, also matching the bound from the Theorem 1$'$.

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
