# OpenReview forum: "Gradient-Free Approaches is a Key to an Efficient Interaction with Markovian Stochasticity"
_NeurIPS.cc/2025/Conference — Submitted to NeurIPS 2025_

### Official Review · Reviewer_qodG · 2025-07-01

**Clarity:** 1
**Significance:** 3
**Originality:** 4
**Rating:** 4
**Confidence:** 2

**Summary:**

This papers studies the finite-sample complexity of zero-order optimization methods with Markovian noise.
It is shown that, in contrast with first-order methods, there are settings where the convergence rate does not depend on the Markov mixing time, but only on the dimension.
This method is then applied for estimating gradients in accelerated gradient descent, providing for a novel method with improved rates.
Corresponding lower bounds are then derived, assessing the optimality of the results.

**Questions:**

1. How is the MLMC method that is proposed to estimate linked with the partitioning of the Markov chain?
2. Important attention is drawn to accelerated gradient descent, but it seems that the most important part of the result is the residual variance term. Wouldn't it be the same for vanilla gradient descent? If not, what would be the result for classical gradient descent, and how could the proposed estimator be used in this setting?
3. How does the proposed method compare, empirically, with classical first-order methods?

**Ethical Concerns:**

["NO or VERY MINOR ethics concerns only"]

**Final Justification:**

I was already positive about this paper, and I remain positive about it. My main concern was on notations that were improperly defined, and the description of the method that was a bit too quick. The authors promised to fix that, which is the reason I remain positive about the paper, and do not decrease my score.

I think the paper should be accepted. Although I am not entirely sure the considered setting is relevant to many practical problems, the results are new and somewhat unexpected, which I believe is of interest to the NeurIPS community.

**Limitations:**

Yes

**Paper Formatting Concerns:**

No issues.

**Quality:**

3

**Strengths And Weaknesses:**

**Strengths**
1. The main strength in this paper is that it shows that, in some zero-order settings (in particular when the dimnesion is larger than the chain's mixing time), the dependency on the chain's mixing time can be removed. I am not excessively familiar with zero-order optimization + Markovian noise, but this seems to be an interesting and original contribution.
2. The approach used to build the gradient is then applied to accelerated gradient descent, showing that the residual variance term in the convergence rate does not depend on the mixing time $\tau$ as long as the dimension $d$ is larger.
3. Corresponding lower bound are provided, matching the residual variance in the one-point and two-point convergence rates that are derived for accelerated SGD.

**Weaknesses**
1. Although the result is interesting in itself, it is not clear whether the mixing time should or should not depend on the dimension. In particular, it may be possible that, when the dimension $d$ grows, $\tau$ also grows relatively to $d$. In such case, there is not real improvement.
2. The paper is hard to read, and notations are not all very well introduced. In particular:
   - significant attention is drawn to the construction of the partitioned noise sequence, which intuitively plays an important role in constructing the proposed estimators. However, these partitioned sequences are never explicitely mentioned after being constructed, which is surprising.
   - the notations J, M, l, B, t, etc. are introduced without any intuitive explanation, which makes it very difficult to understand what idea is trying to be conveyed by the paper.
3. While experiments properly show that zero-shot methods indeed depend on the sum $d + \tau$ of the dimension and the mixing time, it is never compared to first-order methods. Since the central claim in the paper is that zero-order methods can improve over first-order methods in large dimensional spaces, it would be interesting to compare the empirical behavior of both type of methods.

The lower bounds are an interesting contribution of the work, but are only very quickly presented, and almost not discussed in the main text. It would be nice if these discussions can be included in the main paper.

---

> ### Author Rebuttal · Authors · 2025-07-31
>
> We thank the reviewer **qodG** for their thoughtful and constructive evaluation of our work. We address the concerns raised below.
>
> > W1 ($\tau$ may grow with $d$)
>
> Thanks for this remark on the interplay between $\tau$ and $d$. First, we would like to highlight that the acceleration $\tau d$ $\to$ $\tau + d$ we achieve is significant *regardless* of the dependence between growth of $\tau$ and $d$. Second, we think these parameters are of independent nature. For example, mixing time of the chain may represent the complexity of the environment from which the data is sampled, while the problem dimension may be the number of parameters in the chosen model.
>
> Another, more concrete example of an application, where the chain is unrelated to the problem dimension, is a token algorithm (see Section 1.1 from [1]). In short, it is a decentralized optimization algorithm, where a number of users each have a function $f_i(x)$: $R^d \to R$ stored locally, and the users can only communicate with their neighbors in some graph $G.$ Our goal is to minimize the average of these functions. To do so, we start a random walk over $G$, where the current vertex is said to receive the "token", representing the most up-to-date version of $x$. Each time the token moves, the receiving vertex updates it according to its $f_i$. This maps to our framework with random walk over $G$ becoming the Markov chain, and the expectation becoming averaging over all users. Thus $\tau$, the mixing time of the chain, depends solely on the topology of the network between users, while $d$ is the dimensionality of the local models.
>
> [1] Even, M. (2023, July). Stochastic gradient descent under Markovian sampling schemes. In International Conference on Machine Learning (pp. 9412-9439). PMLR.
>
> > W2 (notations $J, M, l, B, t$)
>
> That’s true, the presentation of MLMC technique was rather abrupt. We kept short definitions in Table 2, and the detailed description of these parameters will be added together with some intuition to the Section 2.1 in the final version, thanks! Here is the explanation draft.
>
> To ease the understanding, let’s start with the case $l = 1, M = \infty, B = 1$ and enumerate baseline estimates as $g_1, g_2, \dots$.
>
> Then, the MLMC estimate will be $g_1$ with prob. $1/2$; $g_1 + (g_3 - g_2)$ with prob. $1/4$;
> $g_1 + (g_4 + g_5 - g_2 - g_3)$ with prob. $1/8$ and so on. The parameter $M$ is the upper bound on the number of estimates used. The parameter $l$ transforms the base estimator into a sequence of $l$ base estimators, effectively stretching everything $l$ times. Finally, $B$ serves as a hyperparameter that can multiplicatively increase $l$.
>
> The parameter $t$ is the step of finite difference scheme (see (1)-(3)). It is not tied to MLMC or other parts of the algorithm and the notation is classical.
>
> We hope that after the revision this part is not going to confuse the reader.
>
> > W2 & Q1 (partitioned sequences usage)
>
> You are right, the partitioned sequences are not used by the algorithm. They only appear in Section 2.1 as a motivation for $\tau + d$ behavior. The algorithm then uses a simpler procedure of choosing a random direction per sample. It also achieves a $\tau + d$ rate as we show in appendix (Lemma 5), but the proof is less intuitive. We are going to explicitly mention this in the final version, thanks.
>
> > W3 & Q3 (empirical comparison with first-order methods)
>
> Zero-order (ZO) and first-order (FO) optimization methods are not directly comparable to each other, as they access different oracles. We conducted additional experiments assuming the following information-theoretic equivalence: one FO oracle call is equivalent to $d$ ZO oracle calls.
>
> Specifically we compared our algorithm against Randomized Accelerated GD of [1] on the exact same quadratic problem as in our original experiments (see Section 3) with $\sigma_2^2 = $1e-3. Here is the average optimization error (initial error 10, all errors presented were multiplied by 1e5 for readability), over 1e4 runs, the number of ZO oracle calls is fixed at 2e3 (thus FO gets fewer iterations).
>
> Zero-order case:
> | $\tau$ \ $d$   |   $1$ |   $5$ |   $50$ |
> |----------------|-------|-------|--------|
> | $1$            |  1.88 |  1.92 |   2.43 |
> | $5$            |  1.96 |  1.99 |   2.51 |
> | $50$           |  2.75 |  2.75 |   3.28 |
>
> First-order case:
> | $\tau$ \ $d$   |   $1$ |   $5$ |   $50$ |
> |----------------|-------|-------|--------|
> | $1$            |  0.61 |  4.73 |   9.22 |
> | $5$            |  0.67 |  4.81 |   9.23 |
> | $50$           |  1.37 |  5.36 |   9.25 |
>
> At $d=1$, there are no random directions to choose from, and the ZO algorithm is effectively equivalent to the FO one, except the true derivative is replaced with finite difference approximation, which explains why our results are worse.
>
> At $d=5$, approximation errors no longer play a significant role, as FO algorithm gets 5 times less iterations, and our algorithm convincingly beats it.
>
> At $d=50$, FO does not escape the linear regime within the allowed budget, while ZO performs dramatically better.
>
> [1] Aleksandr Beznosikov, Sergey Samsonov, Marina Sheshukova, Alexander Gasnikov, Alexey Naumov, and Eric Moulines. First order methods with markovian noise: from acceleration to variational inequalities. Advances in Neural Information Processing Systems, 36, 2024.
>
> > S&W (lower bounds almost not discussed)
>
> It was infeasible to keep everything in the main part, so we decided to move the discussion of lower bounds to Appendix F as additional content. In Section 3.2 we present main lower bounds and refer the reader to the appendix for details. If space permits, we will move the discussion into the main text.
>
> > Q2 (vanilla GD performance)
>
> We are not aware of any existing analysis of vanilla SGD in a Markovian setting, but it is expected to have the same stochastic term if used with our gradient estimator, while the linear convergence would be worse. In two-point setting, for example, the expected rate is
>
> $$
>     d\frac{L}{\mu}\log\frac{1}{\varepsilon} + \frac{(d + \tau)\sigma^2}{\mu^2\varepsilon}
> $$
>
> However, the accelerated rate we achieve is important both theoretically to match deterministic lower bounds (see remark on line 295) and in practice, where accelerated methods are dominant (Momentum SGD, Nesterov SGD, Adam).

---

> > ### Comment · Reviewer_qodG · 2025-08-03
> >
> > I would like to thank the authors for their answer to my comments, and I am happy that they consider putting more emphasis on defining the notations and describing the method they propose.
> >
> > I remain overall positive about this paper, provided that the authors include the additional comments from their rebuttal to the final manuscript.

---

> > > ### Author Response · Authors · 2025-08-04
> > >
> > > We appreciate your positive feedback on our work.
> > >
> > > We will indeed incorporate the suggested revisions, including clearer notation definitions and a more detailed method description, into the final manuscript.

---

### Official Review · Reviewer_Ryzo · 2025-07-02

**Clarity:** 2
**Significance:** 3
**Originality:** 3
**Rating:** 4
**Confidence:** 4

**Summary:**

The submission is concerned with zeroth-order (ZO) stochastic optimization in problems where the measurement noise is modeled as a Markov chain. Various settings are considered by the authors (strongly convex smooth, non-smooth). To solve this problem they propose an algorithm combining the optimization techniques for Markovian noise developed in [5] with basic zeroth-order gradient estimators built from samples of function values. Bounds on iteration complexity of are derived for the algorithms. In $d$-dimensional spaces and for Markov chains with mixing time $\tau$, the rates of convergence exhibit a surprising $(d+\tau)$ factor where $d\tau$ was expected, thus claiming that gradient-free optimization may be more efficient than first-order optimization for solving such problems. Numerical experiments are provided to illustrate the theoretical claims.

[5] Aleksandr Beznosikov, Sergey Samsonov, Marina Sheshukova, Alexander Gasnikov, Alexey Naumov, and Eric Moulines. First order methods with markovian noise: from acceleration to variational inequalities. Advances in Neural Information Processing Systems, 36, 2024.

**Questions:**

I did not understand the intuition for the variance reduction on page 6, which yields the factor $(d+\tau) = d  (\tau/d + 1) \geq d \ \text{max}(\tau/d , 1) \simeq d \ \lceil \tau/d  \rceil$, giving just $\tau$ for the variance when $\tau>d$, instead of $d\tau$. I agree that building a minibatch from a subsequence of function samples picked every $d$ successive samples will reduce the mixing time (and the variance of the estimator built on the subsequence) by a factor $\lceil \tau/d  \rceil / \tau$, but if such estimators are derived from $d$ interwined subsequences of $d$ samples, then they form a Markov chain with mixing time $\sim \ \tau$, so I would expect the factor in the variance to scale to $d\tau$ and not $d \lceil \tau/d  \rceil$. Could the authors please clarify this matter?

Detailed comments:

1. On line 84: I think the $\sqrt{d}$ factor in the smoothness coefficient is typical of ZO gradient estimators using 2d samples. How many function samples are required to get this result?

2. In (4), what is the unknown probability distribution $\pi$? I would expected it to be the invariant measure of the Markov chain $\lbrace Z_k \rbrace$. Less specifically, it was difficult to understand the probabilistic setting from the listing in Table 1. I would suggest proceeding as in [5] for reference.

3. In Assumption 2, strong convexity does not imply differentiability. The function $f$ is continuously differentiable and $\mu$-strongly convex, i.e., etc.

4. In Assumption 3, is $|| \cdots ||_{TV}$ the total variation distance?

5. In Assumptions 4 and 4', the squares look quite unnecessary. I believe those conditions derive from the strong growth condition in [5], with the proportional term removed.

6. In Lemma 1, is the hidden factor $c$ in $\preceq$ the same for all $n$, $x$ and $\xi$? I was hoping to find the answer in the proof of Lemma 1, or Lemma 3, but I think the derivation is only available in [5].

7. In (8), the notations $\mathbb{E} _e$ and $\mathbb{E} _Z$ are not consistent with  $\mathbb{E} _\xi$

8. On line 204: $E$ -> $\mathbb{E}$, and I think there is a factor $e$ missing in the two other members of the equation.

9. In (8), there is a factor $e$ missing in the last expression, and I belive that the quantity $|| \hat{g} _{mb} -  \nabla f||$ is controllable only if $\hat{g} _{mb}$ is a gradient estimator using $n$ (or $2n$) function samples.

10. On Line 244, who decompose -> which decomposes

11. On Lines 250 and 251, $|| \dots^2 ||$ -> $|| \dots ||^2$

12. On Line 529: a formal statements -> a formal statement

**Ethical Concerns:**

["NO or VERY MINOR ethics concerns only"]

**Final Justification:**

After the rebuttal and the discussions, I am much more positive about this paper. My comments have been updated in more detail in the section 'Strengths And Weaknesses'.

**Limitations:**

yes

**Paper Formatting Concerns:**

I did not notice any formatting issue.

**Quality:**

4

**Strengths And Weaknesses:**

The submission draws inspiration from a recent article [5] on first-order optimization with Markovian noise. If the concepts and techniques used by the two papers are quite similar, there is a strong contrast in the way the two studies were conducted, and it is apparent that the present submission does not match the original work in terms of consistency, readability, or overall quality. Despite the evident effort involved in producing this paper, I feel unable to give a proper evaluation of the content as I was confused by most discussions and developments. Several results in the main text are just approximations based on intuition, and in particular I did not understand the rationale leading to the $(d+\tau)$ factors in the convergence rates. In my view, the submission does not meet the expected standard of quality for presentation at the conference.

---

**Update after rebuttal**

The authors have committed to improve the presentation of the main text of the submission.

My feelings about this paper have significantly improved after a closer reading of the results reported in the appendix, which helped me better understand the study and acknowledge the merits and contributions of this work.

*Strengths:*

- The main message of the paper (that Markov noise and zeroth-order optimization combine efficiently) is new, interesting, and arguably unexpected.

- The study is meant to be comprehensive. Various settings (strongly convex smooth, non-smooth) and two types of gradient estimators (using one point or two points) are considered. In addition to upper convergence bounds (Theorems 1 and 1'), lower bounds are also derived for the problem (Theorem 2).

- The developments are technically sound. Some of them (Appendices D and E) build on previous results on zeroth-order optimization and in first-order optimization with Markovian noise (which did not make them straightforward). Others (e.g. the lower bound of Theorem 8) are quite inventive.

- The discussions about the performances in various regimes (high correlation, high dimensional) are rich. The numerical experiments are not extensive, but they illustrate well and convincingly the technical results of the paper.

*Weakness:*

- Some results, notions, and notations could have been introduced with more care for easier readability.

---

> ### Author Rebuttal · Authors · 2025-07-31
>
> Many thanks to Reviewer **Ryzo** for their time, detailed review, and suggestions. We now turn to their specific concerns.
>
> > W (several results in the main text are just approximations based on intuition, and in particular I did not understand the rationale leading to the (d + tau) factors in the convergence rates)
>
>
> While the main text emphasizes intuition to aid accessibility, all results — including those involving the $(d + \\tau)$ factors — are rigorously proved in the appendix with appropriate references in the main text. We intentionally placed technical details there to keep the main exposition focused. The $(d + \\tau)$ term is in fact central to our theoretical contribution, and no claims rely on unproven approximations. To address this particular misunderstanding, we note that the partitioned sequences we used to motivate the $(d + \tau)$ behavior are not utilized by the algorithm. The algorithm uses $g\_{rd}$, which is also introduced in the main part with some motivation, but a rigorous proof of its properties is available in Lemma 5 in Appendix (also see the Appendix C Notations and Definitions, as the notation is shortened).
>
> Below, we clarify the notation and expand on the intuition behind this result to make the rationale more transparent. We hope this addresses the reviewer’s concern and helps in evaluating the contribution more fully.
>
> > Q0 (subsequence estimators dependence)
>
> That is true that the resulting d estimators derived from $d$ intertwined subsequences are not independent of each other, thus trivially averaging them would not reduce the variance $d$-fold. However, as these $d$ subsequences each estimate a partial derivative [line 231] *along its separate direction*, we claim the reduction by restoring all $d$ coordinates of the gradient [lines 232-233] with the same accuracy as the naive estimator restored just one.
>
> Let us present this explanation more formally:
>
> The baseline estimator $g$ was able to produce, given a point $x$ and direction $e$, an estimate $g(x, Z, e) = p(x, Z, e) \cdot e$ with variance $\\mathbb{E}\_{e, Z} \\|g - \\nabla f\\|^2$ scaling as $\\frac{D\\tau}{n}$, where $n$ is the number of samples used and $D$ is the baseline variance factor (we omit the explicit value of $D$ here, as we focus on the acceleration only).
>
> Denote by $e\_1, \\dots e\_d$ a random basis - the columns of a random rotation matrix $U$ sampled from the uniform (Haar) measure. First, we construct $d$ new estimators $g\_1, \\dots, g\_d$, each working on its own subchain with mixing time $\lceil \\frac{\\tau}{d}\rceil$, along its own direction ($g\_i = p\_i \cdot e\_i$), and using $\\frac{1}{d}$-fraction of all $n$ input samples. As you mentioned in the question, the variance of each of these estimators will scale as $\\frac{D \\lceil \\frac{\\tau}{d} \\rceil}{n/d} \sim \\frac{D\\max(d, \\tau)}{n}$.
>
> Now, keeping in mind that these estimators may be dependent, we analyze the variance of the final estimate $$g\_{final} = \\frac{g\_1 + \\dots + g\_d}{d}:$$
>
> $$
>     \\mathbb{E}\_{U, Z} \\|g\_{final}(x, Z) - \\nabla f\\|^2 = \\sum\\limits\_{i=1}^{d} \\mathbb{E}\_{U, Z} [ \\langle g\_{final}, e\_i \\rangle - \\langle \\nabla f, e\_i \\rangle]^2 =  \\sum\\limits\_{i=1}^{d} \\mathbb{E}\_{e\_i, Z} [\\frac{p\_i}{d} - \\langle \\nabla f, e\_i \\rangle]^2  = $$
> $$= d \\mathbb{E}\_{e\_1, Z} [\\frac{p\_1}{d} - \\langle \\nabla f, e\_1 \\rangle]^2 = \\frac{1}{d}\\mathbb{E}\_{e\_1, Z} [p\_1 - d\\langle \\nabla f, e\_1 \\rangle]^2 = \\frac{1}{d}\\mathbb{E}\_{e\_1, Z}[ p\_1^2 + d^2\\langle \\nabla f, e\_1 \\rangle^2 - 2d p\_1\\langle \\nabla f, e\_1 \\rangle] = $$
> $$ = \\frac{1}{d}\\mathbb{E}\_{e\_1, Z} \\|g\_1\\|^2 + \\|\\nabla f\\|^2 - 2\\mathbb{E}\_{e\_1, Z} \\langle g\_1, \\nabla f \\rangle
> $$
> As we see, while averaging we didn’t require the independence of $\\{g\_i\\}$, as orthogonal directions always have zero correlation. The final expression is very similar to the $\\mathbb{E}\_{e\_1, Z} \\|g\_1(x, Z) - \\nabla f\\|^2$ that we know scales as $\\frac{D\\max(d, \\tau)}{n}$, but the main term $\\mathbb{E}\_{e\_1, Z} \\|g\_1\\|^2$ experiences additional $\\frac{1}{d}$ factor. We show that even after $d$-fold reduction, it is still a dominant term:
>
> $$
>     \\mathbb{E} g = \\nabla f \\Rightarrow  \\mathbb{E} p \\langle e, \\nabla f \\rangle = \\|\\nabla f\\|^2
> $$
> $$
> [\\mathbb{E} p \\langle e, \\nabla f \\rangle]^2 \\leq \\mathbb{E} p^2\\mathbb{E} \\langle e, \\nabla f \\rangle^2 = \\frac{1}{d}\\|\\nabla f\\|^2\\mathbb{E} \\|g\\|^2 \Rightarrow
> \\mathbb{E} \\|g\\|^2 \\geq d \\|\\nabla f\\|^2
> $$
>
> Therefore the final reduction is $\\frac{\\max(\\tau, d)}{d\\tau} = \\min(d, \\tau)$.
>
> > Q1 ($f\_t$)
>
> We are not sure about the requirement on samples you mention, as there are no estimators presented near the line 84. The function $f\_t$ is a purely theoretic concept with a couple of well known properties: ball smoothing of a Lipschitz function produces a smooth function; finite difference is an unbiased estimator of smoothed function’s gradient [22]. The function $f\_t$ is called the ”smoothed” version of the function $f$. The function $f\_t$ is only a virtual/theoretical object  with good properties. Our algorithm does not use it in any way, but it will be used in the theoretical analysis.
>
> [22] Alexander Gasnikov, Anton Novitskii, Vasilii Novitskii, Farshed Abdukhakimov, Dmitry Kamzolov, Aleksandr Beznosikov, Martin Takac, Pavel Dvurechensky, and Bin Gu. The power of first-order smooth optimization for black-box non-smooth problems. Proceedings of the 39th International Conference on Machine Learning, 2022.
>
>  > Q2 (\\pi in problem definition)
>
> It is indeed the chain's invariant distribution [Line 170]. We will add a reference to Assumption 3 to the problem definition (4).
>
> > Q4 (total variation)
>
> Yes, it is a total variation distance. We will add it to the Notations & Definitions section, thanks.
>
> > Q5 (squares in noise assumptions)
>
> We kept squares to stay in line with other similar assumptions, which require squares either because there is an expectation on LHS, or there is an additional $\\delta^2$ on RHS.
>
> > Q6 (definition of $\\lesssim$)
>
> $\\lesssim$ is defined in Table 1 as problem independent, thus the hidden factor is the same. We agree that “problem independent” is not ideal phrasing and will replace it with “independent of all other variables” / “universal”.
>
> > Q9 (minibatch variance lower bound)
>
> Thanks for the detailed reading of this part, we fixed the missing $e$, as well as added one more transitional step in (8).
> Here is how the section now looks like:
>
> In this section, we describe the main tools used to establish the $(d + \\tau)$-type scaling of the error rate. We will focus on reducing the variance and bias of gradient estimators using a specialized batching approach.
>
> We begin by fixing a common building block of our gradient estimators at a point $x$ for both one-point and two-point feedback, as introduced in Section1.1:
> $$
>     \\hat{g}(x, Z^{(\\pm)}, e) = d \\cdot p(x, Z^{(\\pm)}, e) \\cdot e = e \\cdot \\begin{cases}
>         d\\dfrac{F(x + te, Z^+) - F(x-te, Z^-)}{2t} & \\text{(one-point),}
>         \\\\
>         d\\dfrac{F(x + te, Z) - F(x-te, Z)}{2t} & \\text{(two-point).}
>     \\end{cases}
> $$
> These estimators exhibit a twofold randomness that affects how rapidly they concentrate around the true gradient, as we will discuss below.
>
> For clarity, we focus our discussion on the one-point case, although our conclusions extend to the two-point case as well.
>
> A widely used variance reduction technique is *mini-batching*, where one computes $F(x, Z\_i)$ over a batch of noise variables $\\{Z\_i\\}\_{i=1}^{n}$. The mini-batch gradient estimator is given by:
> $$
>     \\hat{g}\_{mb}(x) = \\frac{1}{n}\\sum\\limits\_{i=1}^{n}\\hat{g}(x, Z\_i^{\\pm}, e) =d\\overbrace{(\\frac{1}{n}\\sum\\limits\_{i=1}^{n}p(x, Z\_i^{\\pm}, e))}^{p\_{mb}} \\cdot e.
> $$
> Let us estimate the scaling of its variance $\\mathbb{E}\_e \\mathbb{E}\_Z \\|\\hat{g}\_{mb} - \\nabla f\\|^2$ with the noise level $\\sigma\_1^2$.
>
> As $\\mathbb{E}\_{Z}\\hat{p}\_{mb} \\approx \\frac{f(x + te) - f(x - te)}{2t} \\approx \\langle\\nabla f, e\\rangle$ we would like to estimate the following for any fixed direction $e$:
> $$
>     \\mathbb{E}\_{Z}\\bigr[p\_{mb}(x) - \\langle\\nabla f, e\\rangle
>     \\bigl]^2
>     \\approx
>     \\frac{1}{t^2}\\mathbb{E}\_{Z}\\Bigl[
>     \\frac{1}{n}\\sum\\limits\_{i=1}^{n} F(x + te, Z\_i^{+}) - f(x + te) \\Bigr]^2
>     \\overset{(1)}{\\approx}
>     \\frac{\\tau}{n}\\frac{\\sigma\_1^2}{t^2} \\,.
> $$
> An important note here is that the last transition assumes a “worst case” scenario, when inequality in Lemma 1 is tight. For example, it is tight for the lazy markov chain discussed below.
> With that, we bound the variance:
> $$
>     \\mathbb{E}\_e \\mathbb{E}\_Z \\|{\\hat{g}\_{mb} - \\nabla f}\\|^2
>     \\geq
>     \\mathbb{E}\_e \\mathbb{E}\_Z \\|{\\hat{g}\_{mb} - \\mathbb{E}\_{Z}\\hat{g}\_{mb}}\\|^2
>     =
>     \\mathbb{E}\_e \\mathbb{E}\_Z \\|{d [p\_{mb} - \\mathbb{E}\_{Z}[p\_{mb}]] \\cdot e}\\|^2
>     =
>     \\\\ \\notag
>     d^2 \\mathbb{E}\_e \\mathbb{E}\_Z | p\_{mb} - \\langle\\nabla f, e\\rangle |^2 \\overset{(7)}{\\approx}
>     \\frac{d^2\\tau\\sigma\_1^2}{nt^2}.
> $$
>
> As for the high-level idea if it is possible to control $\\|\\hat{g}\_{mb} - \\nabla f\\|$, we stress that $\\nabla f$ vanishes after the first step in (8), as we provide *lower* bound for variance. We also added the following remark:
>
> We provide bounds in a “worst case” scenario, when inequality in Lemma 1 is tight. For example, it is tight for the lazy markov chain discussed below.
>
> > Q 3, 7, 8, 10, 11, 12
>
> We fixed it, thank you!

---

> > ### Comment · Reviewer_Ryzo · 2025-08-05
> >
> > Thank you very much for your detailed answers.
> >
> > About Q0 (subsequence estimators dependence): Thanks for the clarification. I confirm that all those derivations appear clearly in the appendix.
> >
> > About Q9: I was confused by the first inequality,
> >
> > $\mathbb{E} _e \mathbb{E} _Z ||{\hat{g} _{mb} - \nabla f}||^2
> >     \geq
> >     \mathbb{E} _e \mathbb{E} _Z ||{\hat{g} _{mb} - \mathbb{E} _{Z}\hat{g} _{mb}}||^2$,
> >
> > and mostly wondering how the one- or two-point gradient estimate (in one direction) could compare to the full gradient. I think the answers are given in Lemmas 6 and 11.
> >
> > About Q1: My mistake, which followed the confusion raised in Q9.
> >
> > Comment after rebuttal: I, for one, still believe the overall presentation of the submission does not fully serve the work behind it, but it may just be me. The discussions of Section 2.1 in particular got me confused. And often it felt useful for me to look up related studies (e.g. [5]) to understand some aspects of the algorithm. However, following your suggestion and in light of the other reviews, I have reconsidered the paper, giving a much closer look at the developments reported in the appendix, which indeed were helpful in better understanding your study and better appreciating the work done. So, I have new comments and questions, hoping there is still time for you to see them. Most comments are possible typos.
> >
> > In my understanding of the developments in Appendices D and E,
> > the factor $(\tau + d)$ follows from the fact that noise correlation and ZO gradient estimation independently affect two different terms in the error variance computation. The result is interesting and positive, and it could not necessarily have been predicted in advance. So I was wondering why a larger factor $\tau d$ was expected here. Could the authors elaborate on this initial (pessimistic) prediction? Why is $(\tau + d)$ surprising to you?
> >
> > a. I struggled to get how the the function samples are collected and arranged into the batches. I understand you use the first $2^{J _1}l$ samples to compute a first estimate $\hat{g} _{ml}(x _0)$, then the next $2^{J _2}l$ to compute the second gradient estimate $\hat{g} _{ml}(x _1)$, etc.? Maybe I missed something here, but It would be useful to indicate how this is done.
> >
> > b. On l. 547: smooth of even differentiable -> smooth or even differentiable
> >
> > c. Is exponent $j$ not missing in (41),(42),(43)? (five occurrences)
> >
> > d. Are there no hidden coefficients missing in (40) and (41) due to [49, Lemma 9]?
> >
> > e. On l.640: is it not $2\mathbb{E} \lbrack ||\hat{g}^j - \tilde{g}^j||^2 + ||\tilde{g}^j - \nabla f _t ||^2  \rbrack$ ?
> >
> > f. I think there is $+$ missing at the top of p.22, on the first line.
> >
> > g. On l.675 and l.705, I believe the conditions on $\Delta$ such as $\epsilon \geq \frac{d\Delta\sqrt{L}}{\mu^{3/2}}$ are quite restrictive as $\epsilon$ is intended to be small. Or is the question of the adversarial bias mostly useful when it comes to clipping as on l.880 in Appendix F.2?
> >
> > h. On l.819, I did not understand the comment on the equivalence of first-order and zero-order information.
> >
> > i. In (70), Assouad's Lemma. Could it be 'max' in place of 'min'?
> >
> > j. On l.841, is the last part in the equation the first member of the equation on l.845?
> >
> > k. On l. 846. This is unimportant, but could there be a coefficient $2k/s^2$ instead of $4k/s^2$ for the KL divergence? I also found another coefficient for the result on l.849.
> >
> > l. On l.851, S(x) -> $S(x)$.
> >
> > m. On l.875, in the equation, is it not $\sigma _1$? I also computed different terms for the equation on l.876. Could you please double-check them?

---

> > > ### Author Response · Authors · 2025-08-06
> > >
> > > We thank you for the reconsideration and detailed comments. A thorough read of the appendix is also very appreciated. Below, we answer the remaining questions.
> > >
> > > > Confusion in Section 2.1 (inequality in Q9); why $\\tau d$ was expected.
> > >
> > > We suspect that the root cause of the confusion here is a misunderstanding of the purpose of Section 2.1 and its structure. Let us break it down:
> > >
> > > 1. The mini-batching part (up to (8)). It is aimed to informally show the reader that the usual mini-batch approach achieves a rate *at least* $\tau d$ for the “worst-case” oracle. This is why we use the inequality $$\\mathbb{E}\_e \\mathbb{E}\_Z \\|{\\hat{g}\_{mb} - \\nabla f}\\|^2 \\geq \\mathbb{E}\_e \\mathbb{E}\_Z \\|{\\hat{g}\_{mb} - \\mathbb{E}\_{Z}\\hat{g} \_{mb}}\\|^2,$$ limiting the $\\mathbb{E}\_e \\mathbb{E}\_Z \\|{\\hat{g}\_{mb} - \\nabla f}\\|^2$ from *below* (it is "$\\geq$"!, the central second moment is less than the second moment). It does not require any additional lemmas ($g\_{mb}$ is not used after this at all) and is not tied to any part of our proofs (Lemma 6 and 11 you mentioned are unrelated, as they provide *upper* bounds for variance). This part is a justification for a) expected $\\tau d$ behaviour b) why our method is not a simple mini-batch approach.
> > >
> > > 2. The “subsequence estimators” part. It is aimed to *informally* show the reader that there may be a space for improvement, motivating the $\\tau + d$ rates.
> > >
> > > 3. Random directions part. Finally, here we present an estimator that is related to our algorithm, and not just a motivation. A sketch of formal proof is also presented.
> > >
> > > 4. MLMC part. The final estimator is presented. We plan to expand this part by providing more explanations to the parameters $J, M, l, B$ and how the $g\_{ml}$ is formed. Here is the sketch:
> > >
> > > To ease the understanding, let’s start with the case $l = 1, M = \\infty, B = 1$ and enumerate baseline estimates as $g_1, g_2, \\dots$.
> > >
> > > Then, the MLMC estimate will be $g_1$ with prob. $1/2$, $g_1 + (g_3 - g_2)$ with prob. $1/4$, $g_1 + (g_4 + g_5 - g_2 - g_3)$ with prob. $1/8$ and so on. The parameter $M$ is the upper bound on the number of estimates used. The parameter $l$ transforms the base estimator into a sequence of $l$ base estimators, effectively stretching everything $l$ times. Finally, $B$ serves as a hyperparameter that can multiplicatively increase $l$.
> > >
> > > We hope that after this discussion Section 2.1 is not going to be confusing. In the final version, we will try to split it into a motivational part and a part that is used by the algorithm & proofs to prevent any misunderstanding.
> > >
> > > > Q a
> > >
> > > You are right, $2^{J_k} l$ samples on iteration $k$. We hope the explanation above will ease the understanding. We are also going to add an extensive discussion of the statistical behaviour of the *total* number of samples used (it appeared to be non-trivial, see **Exow**’s Q4 and our answer for details).
> > >
> > > > Q d
> > >
> > > Right, it should be $\lesssim$, not just $\leq$.
> > >
> > > > Q g
> > >
> > > Thank you for this question. We think the following remark should help:
> > >
> > > In contrast to stochastic problems when one can guarantee a monotone convergence to the optima, the adversarial noise experiences a different, threshold behavior [1]. This is also the case in our result - the rate achieved by the algorithm does not depend on $\\Delta$ as long as it is greater than a certain threshold. Furthermore, the maximum admissible level of noise for our algorithm $\\Delta \\lesssim \\frac{\\varepsilon\\mu^{3/2}}{d\\sqrt{L}}$ matches the minimax optimal bound attained in [1] for general convex $1$-Lipschitz functions and small errors $\\Delta \lesssim \frac{\\varepsilon}{d}$.
> > >
> > > Therefore, the condition on $\\Delta$ is limiting, but it is a typical behaviour for adversarial bias. We will include this remark, thank you.
> > >
> > > On l.880, the use of adversarial bias helps to elegantly finish the proof, but can be avoided through some technical work outlined on l.881-885.
> > >
> > > [1] Andrej Risteski and Yuanzhi Li. Algorithms and matching lower bounds for approximately-convex optimization. Advances in Neural Information Processing Systems, 29, 2016.
> > >
> > > > Q h
> > >
> > > For a $1$-dimensional quadratic problem $F(x, z) = (x - z)^2$ considered in [5], the first-order oracle $\\nabla F(x, z) = (x - z)$ and the zero-order oracle $F(x, z) = (x - z)^2$ are essentially the same, revealing the (scholar) value of $z$ on each call. This allows us to transfer their lower bound to zero-order unchanged.
> > >
> > > > Q j
> > >
> > > Yes, it is the same equation with a big comment in between. We will fix the formatting.
> > >
> > > > Q k
> > >
> > > $\\frac{2k}{s^2}$ it is, thanks! We will fix the arithmetic on l.849 accordingly.
> > >
> > > > Q m
> > >
> > > l.875: Yes, it is $\\sigma\_{1}$.
> > >
> > > L.876: You are right, we forgot to square the $t$ with $\sigma_1$. Therefore, the final variance inflation is $t^2$ times and not just $t$ times. Since $t$ is logarithmic in $N$, it doesn’t change much. But thanks, we will fix it.
> > >
> > > > Q b, c, e, f, i, l
> > >
> > > Thanks, fixed!

---

> > > > ### Comment · Reviewer_Ryzo · 2025-08-07
> > > >
> > > > Many thanks for all these clarifications. I appreciate your commitment to refining the presentation of the main text. I will fix my review shortly (positively).

---

> > > > > ### Author Response · Authors · 2025-08-08
> > > > >
> > > > > We sincerely thank you for the fruitful discussion and we are glad that your final review is positive.
> > > > >
> > > > > Your responsible and careful reading of the article greatly improved the quality of the presentation.

---

### Official Review · Reviewer_xojL · 2025-07-02

**Clarity:** 3
**Significance:** 3
**Originality:** 3
**Rating:** 5
**Confidence:** 3

**Summary:**

The manuscript studies zeroth-order stochastic optimisation when only Markovian noisy function evaluations are available. Its core contribution is a multi-level Monte-Carlo (MLMC) gradient estimator embedded in an accelerated Nesterov-style scheme (Algorithm 1, “Randomised Accelerated ZO-GD”) that works under both one- and two-point feedback and for smooth and non-smooth objectives .
Theoretical results show that, with suitable hyper-parameters, the method attains $\tilde O\Bigl(\bigl[1+\tfrac{d}{B}\bigr]\sqrt{L/\mu}\log\tfrac1\epsilon+ \tfrac{(d+\tau)\sigma_2^2}{B\mu^{2}\epsilon}\Bigr)$ oracle calls in the smooth two-point case (Theorem 1) and comparable bounds for one-point or non-smooth settings (Theorem 3).

**Questions:**

1) Could the technique work under a Polyak-Łojasiewicz or weak convexity assumption?
2) The assumptions 4 and 4' induce almost sure bounds on the centred gradient and function value estimators? Can it be weakened?

**Ethical Concerns:**

["NO or VERY MINOR ethics concerns only"]

**Final Justification:**

The author’s reply fully resolves my concerns; I’m keeping my current rating.

**Limitations:**

Yes

**Paper Formatting Concerns:**

None.

**Quality:**

4

**Strengths And Weaknesses:**

Strength:

1) They present the first analysis of an accelerated (Nesterov-style) zeroth-order SGD algorithm that works with both one-point and two-point feedback when the oracle noise is generated by a Markov process.It achieves the oracle complexity that scales only with $d + \tau$ (dimension $d$ and mixing time $\tau$) instead of the previously conjectured multiplicative factor $d\tau$. When $\tau ≤ d$ the rate matches the i.i.d.-noise case, showing a genuine acceleration.
2) They extend it to non-smooth objectives via a standard smoothing trick, the same framework yields comparable upper bounds in the non-smooth setting, again reflecting the favourable $d+\tau$ dependence.
3) Coupling the estimator with Nesterov momentum yields the familiar $\sqrt{L/\mu}$ dependence while retaining zeroth-order robustness.
4) The authors prove the first lower bounds for Markovian optimisation with zeroth-order feedback (one- and two-point). These bounds match the algorithm’s guarantees up to log factors, demonstrating that the obtained rates are essentially tight.

Weaknesses:
1)  Oracle complexity depends on $\tau$ but users rarely know the chain’s mixing rate.
2) The constants $L,\mu,\sigma,\Delta$ must be a-priori; no adaptive or data-driven estimation is provided.

---

> ### Author Rebuttal · Authors · 2025-07-31
>
> Thanks to Reviewer **xojL** for their evaluation. Our responses follow.
>
> > W1,2 (parameters must be a-priori)
>
> While we agree that this fact limits applying our algorithm in practice, and adaptive algorithms are an interesting direction of research, we feel that this limitation is ubiquitous in optimization and our theoretical contribution is significant enough on its own. We will propose resolving this issue as future work.
>
> > Q1 (weakened convexity assumptions)
>
> While extending our result to other settings is obviously beneficial, we do not aim to cover all of them in this paper. This is because our main contribution is not the $(\\tau + d)$ oracle complexity in one regime, but rather the gradient estimate scheme with reduced variance that allows us to do that. This estimate is not tied to the strong convexity and we think it can be used with almost no changes in numerous other Markov ZOO problems, including nkXR’s suggestions (convex, non-convex) and Polyak-Łojasiewicz or weak convexity assumptions.
>
> > Q2 (weakened noise assumptions)
>
> As long as Lemma 1 is true the proof should work the same. Investigating the sharp theoretical limits for the proposed proof scheme to work is an interesting direction and “Ergodic mirror descent” paper [14] suggests both Markovianity and uniform boundness of the oracle can be relaxed. Nevertheless, we argue that presenting the first result of this type in our limiting but clear setting is beneficial, as the main insights become easier to understand.

---

### Official Review · Reviewer_EXow · 2025-07-04

**Clarity:** 2
**Significance:** 3
**Originality:** 3
**Rating:** 5
**Confidence:** 4

**Summary:**

The paper studies the zeroth-order stochastic optimization problem with Markovian data. The key technical ingredient is a new observation: by using different directions for function value queries along the Markov trajectory, the batched stochastic gradient noise variance can be reduced from $d^2 \tau$ to $d (d + \tau)$ under one-point oracles, and from $d \tau$ to $d + \tau$ under two-point oracles ($d$ is the problem dimension, and $\tau$ is the mixing time). Combining this observation with multi-level Monte Carlo and accelerated gradient methods, the paper proves improved query complexity bounds for the new algorithm. Matching minimax lower bounds are also presented.

**Questions:**

- The choice of the parameter $p$ is not explained and hidden deeply in the appendices. (Even in appendix, it is described only in parenthesis in the theorem statements and difficult to find). Since this parameter is important and affect other parameters $(\beta, \eta, \theta, M)$. This needs to be clarified.

- It seems according to Theorem 1 and 1' that there are no reasons to choose $B$ larger than 1. If so, why keep this parameter?

- The description of the MLMC procedure is unclear. First, it seems to me that the batch size is $B \cdot \max (2^J \ell,  M)$ instead of $2^J \ell$. Second, when writing $\hat{g}[\ell]$, it is not always clear which subset of the total dataset is used.

- The batched sample size $2^J \ell$ could lead to heavy-tailed behavior, when $J$ follows a geometric distribution. While the paper analyzes the expected number of oracle calls, it would be better to have a high-probability bound. (especially when $M$ is large, and the gap between expectation and worst-case upper bound becomes large).

- I do not understand the row for $\eta$ in Table 2.

**Ethical Concerns:**

["NO or VERY MINOR ethics concerns only"]

**Final Justification:**

I have read the rebuttal. I think this will be a good and solid technical paper after improving the presentation based on my comments. I will keep my recommendation.

**Limitations:**

The paper addressed limitations of the work, and the paper does not involve any potential negative societal impact.

**Paper Formatting Concerns:**

No formatting concerns.

**Quality:**

4

**Strengths And Weaknesses:**

Strength: The results are solid and improve upon existing zeroth-order optimization literature. The key observation is clean and intuitive, and the paper made good efforts in explaining the idea in Section 2.1. The results are also complete and comprehensive, covering one-point and two-point oracles, smooth and non-smooth cases, and lower bounds are also established. Overall, the paper presents good technical contribution.

Weakness: Some parts of the presentation, especially those associated to the algorithm details, can be improved for better readability.

---

> ### Author Rebuttal · Authors · 2025-07-31
>
> We thank Reviewer **Exow** for their time, work and positive feedback. Next, we answer the questions raised.
>
> > Q4 (upper bound on oracle calls)
>
> Thanks for this question, it turned out to be insightful. We will add the discussion below to the final version of the paper.
>
> The number of iterations $N$ to achieve the accuracy $\\varepsilon$ in expectation is deterministic. The number of oracle calls, however, is random, as the batch size $b_i$ on step $i$ comes from the truncated “log-geometric” distribution
> $$
>     b_i = \\begin{cases} 2^{J_i} l, & 2^{J_i}  < M \\\\ l, & \\text{else} \\end{cases}, \\quad J_i \\sim \\text{Geom}(1/2)
> $$
> We note here that $l \\sim \\log M$ and $\\mathbb{E} b_i \\sim l\\log M$ are (poly)logarithmic in $M$, but the variance is large, $\\mathbb{E} b_i^2 \\simeq l^2M$. Luckily, the total number of oracle calls $S_N = \\sum\\limits_{i=1}^{N} b_i$ is a finite sum of a large number of such i.i.d. variables and we may hope that the mean value of the sum will be typical (in the sense of high probability, similar to LLN or CLT). The main complication is that $M$ may depend on $N$, thus blocking the use of CLT. Any further discussion requires knowing the dependence between $N$ and $M$, and it can be recovered from the definition of $M$ and stepsize tuning lemma (Lemma 10). For big enough $N$, we can bound $M \\lesssim \\frac{N}{\\log N}$. With this, we apply the finite-time concentration inequality (Bernstein inequality) to our sum:
>
> $$
>     P(S_N > \\alpha \\mathbb{E}[S_N]) \\leq \\exp(-\\frac{\\alpha^2N^2 [\\mathbb{E}b_1]^2}{2N \\mathbb{E}[b_1^2] + \\frac{2\\alpha}{3}MN[\\mathbb{E}b_1]}) \\leq \\exp({-c\\frac{\\alpha^2 N^2 l^2 (\\log M)^2}{N M l^2 + \\alpha M N l(\\log M)}}) \\leq e^{-c (\\log M)^2 \\alpha}.
> $$
>
> It shows the subexponential behaviour of the normalized deviation from the mean, thus confirming that the expectation is typical in the high-probability sense.
>
> An important note here is that the normalization of error is of order $N$, and not $\\sqrt{N}$ as in CLT. Our attempt to use the Berry-Esseen type of results failed, as $\\mathbb{E}b_i \\gtrsim M^2$ is much larger than required $\\sqrt{N}$.
>
> > Q1 (choice of $p$ is obscured)
>
> Agreed, thank you. In the final version we will clarify that $p$ either does not depend on the problem at all or $\\simeq B / (B + d)$, depending on the setting (one-point or two-point).
>
> > Q2 (parameter B is useless)
>
> Fair remark, B overcomplicates understanding the presented results. Our goal was to highlight that the algorithm benefits (in terms of iteration complexity) from having bigger than necessary batches, which was condensed into remark on optimal iteration complexity (line 295). We will add a separate, easier to read corollary in the final version, with $B = 1$.
>
> > Q3 (formula for batch size)
>
> True, and it is misrepresented by line 264 and the table of parameters. Batch size is $ 2^J l$, if $2^J \\leq M$, else $l$. Will be fixed, thanks!
>
> We also plan to add more explanations to the MLMC procedure, here is the draft:
>
> Let’s start with the case $l = 1, M = \\infty, B = 1$ and enumerate baseline estimates as $g_1, g_2, \\dots$.
>
> Then, the MLMC estimate will be $g_1$ with prob. $1/2$, $g_1 + (g_3 - g_2)$ with prob. $1/4$,
> $g_1 + (g_4 + g_5 - g_2 - g_3)$ with prob. $1/8$ and so on. The parameter $M$ is the upper bound on the number of estimates used. The parameter $l$ transforms the base estimator into a sequence of $l$ base estimators, effectively stretching everything $l$ times. Finally, $B$ serves as a hyperparameter that can multiplicatively increase $l$.
>
> > Q3 (notation $\\hat{g}[l]$)
>
> This is a slight abuse of notation of $\\hat{g}_{rd}$, with the rationale that definition would be too bulky if written completely straight. We think the situation could be improved with the explanation above.
>
> > Q5 (definition of $\\eta$)
>
> The two terms in this row are equal and both can be used as a definition of $\\eta$. We will clarify this, thanks.

---

> > ### Comment · Reviewer_EXow · 2025-08-07
> > **post-rebuttal comment**
> >
> > Thank the authors for writing the response. My comments are adequately addressed. I think this will be a good paper after some presentation adjustment. I will keep my score.

---

> > > ### Author Response · Authors · 2025-08-08
> > >
> > > We are grateful for your work, helpful comments and high evaluation of the paper.

---

### Official Review · Reviewer_nkXR · 2025-07-05

**Clarity:** 3
**Significance:** 3
**Originality:** 3
**Rating:** 5
**Confidence:** 4

**Summary:**

This paper investigates zero-order optimization of objectives of the form
$$f(x) = \mathbf{E}[F(x; Z)]$$
where $F$ is convex in its first argument (in fact, $f$ is strongly convex and smooth in this paper). The authors show that in the case of Markovian noise with mixing time $\tau$, it is possible to develop zeroth-order methods whose convergence (relative to standard first-order methods) is worse by a factor of $d + \tau$, which is sharper than a naive $d\tau$ analysis would give. (Here, I focus on the results for two-point feedback; the story is similar for one-point feedback.)

**Questions:**

1. Do the authors truly require the noise to be Markovian, or is it enough that it is (say) $\phi$ or $\beta$ mixing? I did not read the proofs carefully--the supporting lemmas on variance were enough to convince me that the results are true, and I don't want perfect to be the enemy of the good in actually finishing this review. I would be surprised if Markov noise is indeed essential. (It isn't, for example, in the paper "Ergodic mirror descent" the authors cite.)

2. I would assume that without strong convexity, but so long as smoothness is present, one can achieve convergence rates (e.g., in the two-point setting) of
$$\sqrt{LR^2 / \epsilon} + (d + \tau) \sigma^2 / \epsilon^2$$
Is this true? Is there a reason the authors chose to work with strong convexity?

3. In the non-smooth case, does all of this fall apart?

4. [Unnecessary to deal with this but for my curiosity]: If we know a problem is strongly convex, it seems a bit unrealistic to believe we cannot compute derivatives. What happens if the problem is non-convex? Do these methods achieve stationary points with similar improved convergence rates?

**Ethical Concerns:**

["NO or VERY MINOR ethics concerns only"]

**Final Justification:**

I was positive about the paper and remain positive about the paper. The authors' answers all make sense to me. One thing is that I think the title could be revised: I'd say a more accurate title would be something like

**The limited impact of non-i.i.d. noise on zeroth-order optimization**

(or perhaps *The limited impact of non-i.i.d. noise on gradient-free optimization* or somehting). This would be more accurate than the current title, and more accurately capture the message of the paper than

> Gradient-Free Approaches is a Key to an Efficient Interaction with Markovian Stochasticity.

I think this is especially true as (i) there is no "key" to gradient-free approaches, (ii) there is nothing Markovian in the paper or noise, and (iii) the current title is just bad grammar.

**Limitations:**

see questions above

**Quality:**

3

**Strengths And Weaknesses:**

The paper is clear and fairly well-written; I was in a bit of a rush so I will give limited feedback there.

The insight that because of the particular structure of zero-order methods, one can adapt procedures so that mixing times only cause an additive rather than multiplicative penalty, is a nice one. I think the authors oversell that it is surprising (I do not like editorializing in papers, nor do I like adjectives), but that's fine. The basic insight is that (roughly) when we consider gradient-type estimators of the form
$$g = \frac{F(x + \epsilon U; Z) - F(x; Z)}{\epsilon} U$$
then, when the random variables $Z$ come from a mixing Markov chain, we can effectively get "free" gradient estimates at new random directions $U$ on new observations $Z$, up to the dimension, because each new random direction $U$ is quite independent of the previous directions (until we get to dimension $d$). This is a nice insight! I think that for smooth functions, related insights have appeared in distributed optimization before, where delays due to communication introduce gradient errors that are asymptotically smaller than the noise inherent in the problem (see, e.g., https://arxiv.org/pdf/1104.5525 or https://arxiv.org/abs/1508.00882). It seems to me that the insights here relate to those.

In any case, it is nice that the paper can demonstrate these speedups so convincingly.

Weaknesses:
See my questions below.

---

> ### Author Rebuttal · Authors · 2025-07-31
>
> We appreciate Reviewer **nkXR**’s time, effort, and supportive feedback. Below, we address their questions.
>
> > Q1 (Markovianity to general mixing)
>
> You are right, we do not use Markovianity at full potential. As long as Lemma 1 is true the proof should work the same. Investigating the sharp theoretical limits for the proposed proof scheme to work is an interesting direction and “Ergodic mirror descent” paper suggests both Markovianity and uniform boundness of the oracle can be relaxed. Nevertheless, we argue that presenting the first result of this type in our limiting but clear setting is beneficial, as the main insights become easier to understand. We are going to add this as a future work, thanks!
>
> > Q2 (strongly convex to general convex)
>
> There is a classical regularization trick that transforms results in strongly convex setting into general convex by adding a quadratic part. For example, given a convex function $g$, we can run the optimization procedure querying the function $f(x) = g(x) + \\frac{\\mu}{2}\\|x\\|^2$ that is strongly convex. The solution produced by the algorithm may be only slightly worse for $g$ than for $f$:
> $$
>     |f(x^N) - f^\*| \\leq \\mu R^2 + |g(x^N) - g^\*|.
> $$
> Choosing $\\mu = \\frac{L}{N^2}$ and plugging in our convergence guarantee, we get the desired result:
> $$
>     \\sqrt{\\frac{LR^2}{\\varepsilon}} + \\frac{(d + \\tau)\\sigma^2R^2}{\\varepsilon^2}
> $$
> There were no special reasons for choosing strong convexity except for personal choice.
>
> > Q3 (non-smooth)
>
> We believe the question was “is it possible to relax strong convexity and smoothness at the same time?”. The same regularization trick should work in the non-smooth setting. If applied to the Theorem 3’  from Appendix B.2, the resulting rate is
> $$
>     \\frac{d^{1/4}G}{\\varepsilon^{1/2}} + \\frac{(d + \\tau)G^2R^2}{\\varepsilon^2}.
> $$
>
> While presenting new corollaries of the result is obviously beneficial, we do not aim to cover all of them in this paper. This is because our main contribution is not the $(\\tau + d)$ oracle complexity in one setting, but rather the gradient estimate scheme with reduced variance that allows us to do that. This estimate is not tied to the strong convexity and we think it can be used with almost no changes in numerous other Markov ZOO problems, including Q4 (non-convex) and **xojL**’s suggestions (Polyak-Łojasiewicz or weak convexity assumptions).
>
> > Q4 (strongly convex to non-convex)
>
> There is no straightforward technique to transfer results from strongly convex setting to non-convex that we are aware of, but we believe the $\\tau + d$ behaviour will remain the same (see answer to the Q3). Particular recent works [1, 2] proposed a method for both non-smooth and non-convex stochastic optimization and may serve ground for the development of our technique.
>
> [1] Guy Kornowski and Ohad Shamir. 2024. An algorithm with optimal dimension-dependence for zero-order nonsmooth nonconvex stochastic optimization. J. Mach. Learn. Res. 25, 1, Article 122 (January 2024).
>
> [2] Yuyang Qiu, Uday Shanbhag, and Farzad Yousefian. Zeroth-Order Methods for Nondifferentiable, Nonconvex, and Hierarchical Federated Optimization, Advances in Neural Information Processing Systems, 36, 2023.
>
> > S&W (related insights in distributed optimization)
>
> This connection to distributed optimization is new, thanks! We agree that delayed gradients behave similar to the Markovian case. Nevertheless, the speedup from “free” gradient estimates appears in the zero-order case, but not in first-order (see our comment on lines 226-228). This suggests the nature of acceleration may be different.

---

> ### Comment · Reviewer_nkXR · 2025-07-31
> **seems reasonable**
>
> The authors' answers all make sense to me.
>
> One thing is that I think the title could be revised: I'd say a more accurate title would be something like
>
> **The limited impact of non-i.i.d. noise on zeroth-order optimization**
>
> (or perhaps *The limited impact of non-i.i.d. noise on gradient-free optimization* or somehting). This would be more accurate than the current title, and more accurately capture the message of the paper than
>
> > Gradient-Free Approaches is a Key to an Efficient Interaction with Markovian Stochasticity.
>
> I think this is especially true as (i) there is no "key" to gradient-free approaches, (ii) there is nothing Markovian in the paper or noise, and (iii) the current title is just bad grammar.

---

### Note · Authors · 2025-08-12

Dear Area Chairs and Reviewers!

Thank you for your work and helpful feedback on improving our paper!

Reviewers __nkXR, EXow, xojL, qodG__ initially positively assessed our paper and retained this opinion after the rebuttals.

Reviewer __Ryzo__ also positively assessed our paper after the rebuttals and promised to change the review to a positive one in the near future ("I will fix my review shortly (positively)"). We are looking forward to it.

Thanks again!

---

### Decision · Program_Chairs · 2025-09-17

**Decision:**

Reject

**Comment:**

# Summary

This paper studies the finite-sample complexity of zeroth-order (gradient-free) stochastic optimization under Markovian noise. The authors consider both smooth and non-smooth objective functions and propose two algorithms that combine multi-level estimators of the gradient of a smoothed version of the objective with acceleration techniques.
These methods yield improved query complexity bounds compared to naïve implementations. In particular, the bounds are comparable to those achieved by first-order methods under similar settings, up to additional multiplicative terms of order $\tau + d$ or $d(\tau + d)$, depending on the specific algorithm, where $\tau$ denotes the mixing time of the Markov chain and $d$ the problem dimension.

The paper also provides matching minimax lower bounds to establish the optimality of the proposed rates and includes simple numerical experiments to support the theoretical findings.

# Recommendation

All reviewers agreed that the paper is an interesting contribution. Following their opinion, I recommend acceptance.
I strongly encourage the authors to incorporate the reviewers' feedback when preparing the final version of the paper.

===

As recently advised by legal counsel, the NeurIPS Foundation is unable to provide services, including the publication of academic articles, involving the technology sector of the Russian Federation’s economy under a sanction order laid out in Executive Order (E.O.) 14024.

Based upon a manual review of institutions, one or more of the authors listed on this paper submission has ties to organizations listed in E.O. 14024. As a result this paper has been identified as falling under this requirement and therefore must not be accepted under E.O. 14024.

This decision may be revisited if all authors on this paper can provide proof that their institutions are not listed under E.O. 14024 to the NeurIPS PC and legal teams before October 2, 2025. Final decisions will be communicated soon after October 2nd. Appeals may be directed to pc2025@neurips.cc.